# Co-Creation and Co-Production of Health Promoting Activities Addressing Older People—A Scoping Review

**DOI:** 10.3390/ijerph192013043

**Published:** 2022-10-11

**Authors:** Anne Seneca Terkelsen, Christian Tolstrup Wester, Gabriel Gulis, Jørgen Jespersen, Pernille Tanggaard Andersen

**Affiliations:** 1Unit for Health Promotion, Department of Public Health, University of Southern Denmark, 6705 Esbjerg Ø., Denmark; 2Unit for Epidemiology, Biostatistics, and Biodemography, Department of Public Health, University of Southern Denmark, 5230 Odense, Denmark

**Keywords:** co-creation, co-production, health promotion, older people

## Abstract

The global population is aging and the promotion of health and well-being for this generation is essential. Co-creative and co-productive practices can be solutions to welfare challenges in local policies. Therefore, this scoping review aimed to understand the extent and type of evidence in relation to the co-creation and co-production of health-promoting activities addressing older people aged 60+ years and to examine the influence of co-creative and co-productive activities on health and well-being, including influential factors for co-creation and co-production. We searched for peer-reviewed and grey literature in ten scientific and five non-scientific databases. From the 2648 studies retrieved, 18 articles were included in this review. Then, an inductive thematic content analysis was applied to the analysis. Three categories related to co-creative and co-productive activities emerged: “Social and physical activities”, “Development of age-friendly environments”, and “Discussions of healthy and active aging”. Facilitating factors for co-creation and co-production were related to the planning and structure of the process and recognition of participants’ time and resources, while the recruitment of participants and their time and resources were the main barriers. Future studies should target co-creative and co-productive interventions to concrete areas and specific sub-groups and be aware of factors influencing a co-creative or co-productive relationship with older people.

## 1. Introduction

The global population is aging. In 2030 one in six people will be 60 years of age or older worldwide [1]. The definition of older people is, according to the United Nations (UN), people aged 60 years, while the World Health Organization (WHO) defines older people in developed world economies as 65 years of age and above [2]. In this scoping review, we use the UN definition of older people (60+ years).

How can these extra life years be spent in good health? Together with member states, the WHO presented a Global Strategy and Action Plan towards Aging and Health for 2016–2020, followed by The United Nations Decade of Healthy Aging 2021–2030 to foster healthy aging and improve the lives of older people, their families, and their communities [3].

The focus on developing policies to improve healthy aging and well-being is increasing in Europe [4]. Cities and municipalities strive to create physical, social, and service environments that are age-friendly [5]. Additionally, the WHO has developed a framework of eight domains of great importance [4] consisting of three clusters: municipal services, the physical environment, and the social environment. The last cluster, social environment, comprises three domains: social participation, social inclusion and non-discrimination, and civic engagement. The goal of interventions in this cluster is to establish socially inclusive environments in which older people can participate to reduce loneliness and to create opportunities to engage and contribute to social and political life.

Evidence shows that social participation and social engagement are associated with several health benefits of both mental and physical nature in older people [6,7,8,9].

A systematic review of the effect of social interventions on social and health-related measures among nursing home residents showed the positive effect of reminiscence therapy, cognitive and support group interventions on depressive measures, quality of life, and loneliness [6], whilst another review found that social participation was associated with better health outcomes in areas such as cognitive function, depression, self-rated health, physical capacity, functional ability and incidence of falls [9].

Participation does, however, come in many forms, according to Aronstein’s Ladder of Citizen Participation [10]. The ladder has eight rungs referring to the degree of citizen power, from manipulation and therapy at the bottom, to partnerships, delegated power, and citizen control at the top. That is, at the bottom rungs, the degree of citizen control is non-existent and, contrastingly, at the top level of participation, citizens have increased power, and shared decisions are made between the public sector and citizens. New models of citizen participation have since replaced Aronstein’s Ladder and have proposed that the process of citizen participation is not static but a process in which participants at the organizational level and lay people both contribute knowledge and experience [11].

With an increase in the elderly population and related welfare challenges in cities and municipalities, the need for engaging citizens in interactions with healthcare professionals and policymakers has become essential in reducing healthcare costs, providing high-quality service, and, at the same time, increasing citizen satisfaction of the services delivered [12,13].

The concepts of co-creation and co-production are considered potential solutions to welfare challenges and have become important concepts in many public policies regarding the economy, efficient use of resources, and citizen satisfaction [14,15].

The two terms are used interchangeably in practice, with co-creation being the newer term, stemming from commercial business [16], and co-production originally established in the public sector in the 1970s to help explain the role of citizens in the production of public services [17,18]. According to Voorberg et al. [13], the terms can be distinguished by the levels of involvement of the citizens: co-creation refers to the involvement of citizens in the initiation and/or design phase, whereas co-production is considered as the involvement of citizens in the implementation phase of a service.

For decades, the involvement of citizens has been brought into focus to improve the collaboration between the public sector and civil society in Denmark. In recent years, there has been an increasing interest in the development of co-creative and co-productive practices in local policies and the healthcare sector [19,20,21]. Literature on co-creation and co-production in relation to the promotion of health and well-being is, however, sparse.

Two separate block searches with various search terms of co-creation and co-production, health promotion, and older people aged 60+ were conducted in the Cochrane Database of Systematic Reviews and JBI Evidence Synthesis in September 2021. The search in the Cochrane Database of Systematic Reviews revealed only one review where the focus on co-creation was sparse and not relevant to our research questions. In the journal JBI Evidence Synthesis, no articles were found to be relevant to the research questions. Based on these initial searches for scientific literature on the subject, we state the need for clarification on what is known from existing literature about the co-creation and co-production of activities in relation to the health and well-being of older people. The aim of this scoping review is thus to explore the co-creative and co-productive health-promotive activities with older people reported in the literature, to investigate how the co-creation and co-production of the activities affect the health and well-being of older people, and to determine which facilitators and barriers exist for the co-creation and co-production of these activities.

## 2. Materials and Methods

As the topic is considered emerging, we considered a scoping review useful. The main purpose of a scoping review is to identify and map available evidence and is particularly useful for determining the breadth of the area and providing an indication of the volume of published and unpublished literature in the field [22,23]. It allows for systematic identifications of available evidence in the study and analysis of possible knowledge gaps.

The proposed scoping review was conducted in accordance with the methodology for scoping reviews proposed by The Joanna Briggs Institute (JBI) [24]. The methodology is based on a framework originally proposed by Arksey and O’Malley in 2005 [22] and further improvements by Levac and colleagues in 2010 [25]. The PRISMA-ScR (Preferred Reporting Items for Systematic reviews and Meta-Analysis extension for Scoping Reviews) checklist was followed to guide the conduct of the scoping review to increase methodological transparency and the comprehension of our research findings [26]. A scoping review protocol was registered at https://figshare.com (accessed on 5 October 2022) to make the research transparent, in line with the national Code of Conduct for Research Integrity [27].

### 2.1. Inclusion Criteria

#### 2.1.1. Participants

In this scoping review, we focused on both sides of the co-creative or co-productive relationship between the public or the private sector and older people. Thus, only studies with participants from each side of the co-creative or co-productive relationship were included in the literature search.

The eligible participants on the other side of the co-creative or co-productive relationship were people aged 60 years or above (according to the UN definition of older people), with the oldest aged 80 years or above.

Literature concerning the co-creation or co-production of activities with patients, people with serious mental impairments such as severe dementia, children, adolescents, or adults under the age of 60 was not considered eligible for inclusion. However, exceptions occurred in the case of two studies in which a few participants were below the age of 60 and the majority were above the age of 60 as these studies were considered relevant to our research question.

#### 2.1.2. Concept

In the scoping review, we define co-creation and co-production “as a joint effort of citizens and public sector professionals in the initiation, planning, design and implementation of public services” [28]. The concepts of co-creation and co-production are related depending on the levels of involvement of the citizens, and therefore, we chose to include both concepts. As the term co-design is sometimes used as a synonym for co-creation [13], we decided to include this term in our search [29].

Our ambition was to search for physical, sedentary, and other social activities in which the public and private sectors collaborated with older people in either the initial planning, design, or implementation of the activity. That included any peer-reviewed literature of both quantitative and qualitative characters. In addition, we searched for grey literature to identify non-indexed reports, policy documents, government documents, and dissertations.

Literature in which other terms for co-creation or co-production were used, except for co-design, was excluded if it focused on only one side of the co-creative or co-productive relationship.

#### 2.1.3. Context

The activities could take place in a community setting or primary or secondary healthcare either online or in person. Community settings could be public indoor or outdoor spaces whereas the co-creation or co-production in primary or secondary healthcare could take place at nursing homes, activity centers in connection with sheltered housing, hospitals, and the like.

### 2.2. Search Strategy

The search strategy was planned and conducted by the first author with guidance from a research librarian and the author group. To locate both published and unpublished literature from the health and social sciences, a three-step search strategy was conducted as recommended in the JBI Manual for Evidence Synthesis [24]. As the first step, a preliminary search strategy was made together with a research librarian at the University of Southern Denmark and an initial search was performed in the databases MEDLINE, Ovid, and the Web of Science to identify articles relevant to the topic. The search was undertaken on the 24 of September 2021 and revealed 179 hits on MEDLINE and 887 hits on the Web of Science.

Secondly, an analysis of the keywords and index terms of the retrieved papers was performed. The analysis identified additional terms to be included in the search. All search terms were then transferred to the following health and sociological databases found relevant for the search of the topic: CINAHL, Cochrane Library, Embase, Idunn, MEDLINE, PsycINFO, Scopus, SocINDEX, SveMed+, and the Web of Science. A fourth search block exploring the term “activity” was initially added to further specify the search in relation to health-promotive activities. However, adding this fourth block restricted the search and left out potentially relevant studies and it was therefore removed.

Thirdly, we searched the reference list of all identified papers included in the review. No authors were contacted for further information.

A search for grey literature was conducted at the database of UC Knowledge, the repository and showcase for knowledge production at the Danish university colleges, and through websites of the Dane Age Association, Danish Health Authority, The National Board of Social Services, and VIVE, The Danish Centre for Social Science Research.

Literature published in English, Scandinavian languages, and German was considered for inclusion since these languages are known by the authors. As the scientific interest in the field of co-production and health policy began to take hold around 2006, literature from 2005 to 2021 was considered eligible for inclusion [12].

### 2.3. Source of Evidence Screening and Selection

Following the search, all identified literature from the scientific databases was collated and uploaded into the reference management tool, EndnoteX9/2018 (Clarivate Analytics, PA, USA), and duplicates were removed. The first author screened titles and abstracts for relevance in relation to the research questions and eliminated studies based on the inclusion and exclusion criteria. Any questions raised about whether to include a study for full-text screening were discussed with the author group. Full-text articles found relevant were transferred into the web-based screening software, Covidence [30], and assessed in detail against the inclusion and exclusion criteria by two independent reviewers. Disagreements between the reviewers were discussed and agreed upon or otherwise a third reviewer was consulted.

### 2.4. Data Extraction

To provide an overview of the characteristics of the literature included and the main findings of each study, two independent authors extracted data from the included articles. Both authors read a minimum of 15 articles and decided which study characteristics and which results were relevant for inclusion. Any disagreements were discussed and resolved.

The extracted data from the literature included: author(s), year of publication, study design, title, country, the aim of the study, participants from either side of the co-creative or co-productive relationship ((1) the public or private sector and (2) older people), and setting. For result analysis, the activities co-created or co-produced, the potential impact on the health and well-being of older people, and possible factors influencing the co-creation or co-production between the public and private sector and the older people were extracted.

### 2.5. Data Analysis and Presentation of Results

As recommended by Levac et al. [25], the analysis of data can be similar to qualitative data analytical techniques. We decided to apply a qualitative content analysis [31] with an inductive approach to identify what health-promotive activities were co-created or co-produced between the public and private sector and older people, how these activities may impact the health and well-being of older people and identify which factors may influence the co-creation or co-production of activities with older people. One author read the included articles repeatedly and identified categories through three steps as recommended by Elo and Kyngäs [31]. In step one, a free line-by-line coding of the findings was conducted, followed by step two, an organization of the codes into categories, and lastly, the third step, an abstraction of the categories.

## 3. Results

### 3.1. Literature Search

The search for relevant literature returned a total of 3707 peer-reviewed studies and 51 records from websites: Cinahl (*n* = 995), Cochrane Library (*n* = 88), Embase (*n* = 227), MEDLINE (*n* = 257), PsycINFO (*n* = 46), Scopus (*n* = 998), SocINDEX (*n* = 60), the Web of Science (*n* = 1036), and UC Knowledge (*n* = 51). A total of 1059 duplicates were removed which left 2648 articles to be screened by title and abstract. Out of these, 200 articles were found eligible for full-text screening; of which, 23 papers could not be retrieved. Among the remaining articles, 160 did not deal with topics concerning the co-creation or co-production of activities with older people aged 60+ and were excluded. Thus, 18 full-text articles were included [32,33,34,35,36,37,38,39,40,41,42,43,44,45,46,47,48,49]; a flow chart illustrating the study selection process can be seen in Figure 1.

### 3.2. Study Characteristics

Of the 18 included scientific articles seven studies were from the United Kingdom. The remaining articles were from Denmark (*n* = 3), Australia (*n* = 2), The Netherlands (*n* = 2), Canada (*n* = 1), Italy (*n* = 1), Sweden (*n* = 1), and Taiwan (*n* = 1). The studies were conducted from 2016–2020. Most of the studies (*n* = 16) were qualitative intervention studies, one study was a randomized controlled trial (RCT), and one was a study protocol. Most studies (*n* = 12) used the term co-creation, one used the term co-production, and the remaining studies (*n* = 5) used the term co-design. Participants at one side of the co-creation or co-production relationship from the public sector (none were mentioned from the private sector) were governmental and municipal staff such as healthcare professionals and local service providers, researchers, and students from various disciplines (health, design, and architecture). The participants on the other side of the co-creation or co-production relationship were described as older people, older adults, seniors, community-dwelling older adults, care home residents, and older citizens with an age range from 60 to 90 years of age. Care facilities provided the setting for four studies, three operated in a community setting, while the remaining settings were described in relation to the cities and countries in which the study took place. In one study, the setting was online where older people co-created social activities through online communication using web technology. Study characteristics are presented in Appendix A, Table A1.

### 3.3. Co-Created or Co-Produced Health Promotive Activities

From the content analysis, we identified three categories describing co-creation or co-production in relation to activities. They are summarized in the following Table 1.

### 3.4. Health Promotive Activities’ Impact on Health and Well-Being

All included studies described a future aim, or an outcome related to the health and well-being of older people: The studies focusing on the social and physical activities co-created or co-produced were all related to the overall promotion of both mental and physical health and well-being. Some studies emphasized the importance of social participation to prevent social isolation and loneliness, create a sense of belonging, and promote an active old age [32,34], while other studies described the promotion of an active old age through both physical and social activity [43] and encouraged people to stay active and promoted independence and dignity [48].

In the co-creation processes related to the development of age-friendly environments, the focus was mainly on the promotion of health, well-being, and quality of life among older people, while some studies described more specific topics such as opportunities for social participation and active aging. In one study, the outcome of the co-creation workshop was the identification of concrete themes to reduce social isolation and facilitate participation and social cohesion through social gatherings and [37], while another study focused on active aging in the local environment by co-designing urban installations to promote movement and social interaction [46].

Regarding the discussion of active and healthy aging, three studies revolved around active aging and strategies to promote active aging, barriers to staying active, and specific activities to do as part of an active daily life [41,44,45], while two articles discussed the meaning of health and well-being in older age and the opportunities and barriers encountered with aging [40,42].

### 3.5. Factors Influencing the Co-Creation and Co-Production of Activities

More than half of the included studies described their experience of the co-creation or co-production process and identified possible facilitating or hindering factors. From the content analysis, three categories of facilitating and hindering factors emerged. They are summarized in Table 2.

Appendix B, Table A2 provides an overview of health-promotive activities co-created or co-produced, the influence of the activities on the health and well-being of older people, and factors facilitating or hindering the co-creation or co-production of activities with older people.

## 4. Discussion

Overall, the base of evidence was limited, including only 18 studies. We identified three ways to make use of co-creation or co-production in relation to activities: (1) the co-creation or co-production of social and physical activities; (2) the development of age-friendly environments, and (3) discussions of healthy and active aging. We also found that the co-creation process could be facilitated by tangible and general factors, factors related to participants, or be hindered by factors such as recruitment of participants, time and resources, or a lack of structure.

### 4.1. Co-Created or Co-Produced Activities

The co-created or co-produced activities in the studies were equally distributed into three main categories including setting up social and physical activities, developing age-friendly environments, or discussions among participants on themes related to healthy and active aging.

Only 7 of the 18 studies described activities in the sense of recreational or leisure activities to promote health and well-being in older people, which was less than expected. As the terms “recreational” and “leisure activities” are often implied in the term “activity”, we expected most studies to deal with social and physical recreational and leisure activities. We, however, discovered that linking this term, “activity”, to the concepts of co-creation and co-production generated studies in which the term “activity” has a broader meaning and is not necessarily understood as the recreational and leisure activities that may affect health and well-being.

Depending on the purpose of the literature search, it may be relevant to conduct a specific search for recreational and leisure activities since these terms are historically linked to health and well-being [51]. However, as our search aimed for mapping the existing literature on all activities co-created or co-produced between older people and the public or private sector in relation to the promotion of health and well-being in older people, this limitation was not relevant.

**Co-creation or co-production of social and physical activities.** Seven studies described an outcome of the study related to health or well-being such as reduced loneliness, better social, emotional, and physical health, or increased quality of life. It was, however, difficult to distinguish whether the health outcomes were a result of the actual social or physical activity, a result of the co-creative or co-productive process, or a combination of both. As mentioned earlier in this paper, we know from the literature that being involved in social activities including physical activities in older adulthood is associated with both better mental and physical health [6,7,8,9]. We also know that co-creation can empower citizens and enhance mutual trust between stakeholders, enhancing social cohesion [52]. Participating in co-creative or co-productive social and physical activities may build more resilient and sustainable solutions for the benefit of both older people and the public sector, but more research is needed to explore the actual benefits of participating in co-creative or co-productive processes on the health and well-being of the individual. This is in line with the paper by Heimburg, DV. and Cluley, V. who stated that more research is needed on how to link health promotion and co-creation to tackle complex health and well-being issues [53].

Co-creation as **a method to develop age-friendly environments** was described in 6 of the 18 studies. This is in line with strategic objective number two in the WHO Global Strategy and Action Plan which focuses on age-friendly environments and stresses the importance of fostering older peoples’ autonomy and engagement and promotes multisectoral action [54]. In the six studies on the development of age-friendly environments, older people were mainly engaged through co-design. That is, a collaboration between researchers with professional backgrounds in the design or architecture on one side and older people on the other side of the co-design relationship. In a traditional design process, the designer is the primary source of creativity, and this power balance must be shifted towards the older people engaging in the co-design process to foster engagement and meet the needs of the older people in the development of the age-friendly environment [55]. The involvement of older people in a design process may be challenging due to age-related health conditions where tailored engagement techniques may be useful [56]. The use of co-design in environmental developments has, however, been practiced in other fields such as co-designing technology solutions with older people as described in the study of smart home technology co-designed with people living with dementia or Parkinson’s disease [57] and in several studies describing the co-design of technology for aging in place [58]. The effect of the co-design solutions on health and well-being are, however, rarely described and must be further investigated.

The first strategic objective of the WHO Global Strategy and Action Plan is the commitment to action on healthy aging in every country [54]. Not as an independent program but as an integral part of health programs and policies. This implies a change in the attitude towards aging and older people. The negative attitudes and assumptions may affect the policies and actions taken to solve challenges related to the aging population. According to the WHO, ways to combat ageism is to create national and regional frameworks for action including new ways to look at older people and aging and discover the roles and needs of older people today [54]. One way to discover what is essential to older people in relation to active aging may be through **discussions of healthy and active aging** such as the identification of older peoples’ needs in relation to aging well [42] or a discussion of how the collaboration between multiple stakeholders can be an important factor for sustainable welfare solutions for older people [45]. Further co-creative discussions are needed to meet the needs of older people in the future.

### 4.2. Health Promotive Activities’ Impact on Health and Well-Being

The effect of the co-creative and co-productive activities on the health and well-being of older people was only touched upon on a general level. Few studies mentioned a specific outcome related to health and well-being caused by the co-creation and co-production of activities. Measuring the effect of co-creation or co-production can be difficult using traditional measurement techniques and evaluation tools, as the effect of co-creation and co-production processes are often intangible [59]. This might explain why the included studies do not specify the direct health effects of co-creation and co-production. It is beyond the scope of this review to further discuss the measurement of co-creation and co-production efforts. We do, however, emphasize that future studies investigate how direct health benefits such as extra healthy life years can be measured in relation to co-creation and co-production.

### 4.3. Facilitating and Hindering Factors for Co-Creation and Co-Production

While looking into the included studies, we identified a broad variety of facilitating and hindering factors for co-creation and co-production. Co-creation and co-production research have traditionally focused on facilitators and barriers separately [60]. In a review by Voorberg et al., the influencing factors for co-creation and co-production are divided into factors that either relate to the organizational side or the citizen side of the co-creation or co-production relationship and they do not differentiate between facilitating and hindering factors because they see the factors as “two sides of the same coin” [13]. Looking at the influential factors on either side of the co-creation or co-production relationship may be helpful in determining which areas need support along the process.

On the organizational side, Voorberg et al. mention the following as the most dominant influential factors: (1) compatibility of public organizations with citizen participation, (2) attitude towards citizen participation, (3) risk-averse administrative culture, and (4) presence of clear incentives for co-creation or co-production [13].

Even though we did not separate the factors influencing the co-creation or co-production from either the organizational or the citizen side, it does become evident that the influencing factors identified are primarily seen as something happening or not happening at the organizational level. This is in line with the review from Voorberg et al. who believe that it is often the public body that is regarded as responsible for the success of the co-creation or co-production [13]. Two of the included studies in our review emphasized the importance of acknowledging participants’ time, competencies, and resources [33,45] and, in addition, seeing citizens as associates as suggested by Voorberg et al. [13]. Other included studies described the importance of having an agreed aim and a realistic scope of the co-creation project [40,44], which is in line with the influential factor 4, the presence of clear incentives for co-creation or co-production, stated by Voorberg et al. [13]. Presenting clear incentives for co-creation or co-production might overcome barriers such as a lack of structure of the co-creation or co-production process as described in the included studies by Leask et al. and Yuan et al. in which they demand a systematic framework and support and coordination of the co-creation and co-production processes [44,49].

On the citizen side, the most important influential factors according to Voorberg et al. are: (1) the personal characteristics of the citizens, (2) a sense of ownership, (3) social capital, and (4) risk aversion or trust in the co-creation initiative [13]. A sense of ownership in the co-creation was found in both the study by Brookfield et al., who emphasized the importance of letting participants control the level and duration of participation [33], and in the study by Leask et al., who noticed that participants perceived greater control and empowerment by collecting and presenting data in the study [44].

Some of the hindering factors found for co-creation or co-production were related to the recruitment of participants. As stated by Voorberg et al., the first influential factors for co-creation and co-production are personal characteristics, which determine whether citizens are willing and able to participate. In general, people with higher educational levels are more willing to participate in co-creation and co-production projects as they are more aware of community needs and are able to express their own needs [13]; this was also the case in the study by Yuan et al. where study participants had a higher educational level than the population [49]. Additionally, in the studies by Fumagalli et al., Hatton et al., and Leask et al., recruitment of participants was found to be difficult, and participants were recruited by snowball sampling [38], self-selection [42], or only male participants were included [44]. We might expect that the recruitment of older people in co-creation and co-production projects may become more difficult as people age due to both the physical and mental limitations associated with aging. However, physical, and mental capacity can vary across this age group and future co-creation and co-production initiatives targeting various groups of older people are needed.

### 4.4. Strengths and Limitations

This scoping review is the first to systematically search for and retrieve literature on the co-creation and co-production of activities in relation to the health and well-being of older people.

We are, however, aware that the review is subject to some limitations. We chose to include both the concept of co-creation and co-production and to define the concepts together “as a joint effort of citizens and public professionals in the initiation, planning and implementation of public services” [28] based on the notion that the concepts are rarely distinguished in scientific papers [13] and to capture the broadest range of literature on the topic. A separate literature search on each concept could have been made to distinguish what is known from literature about co-creation and co-production in relation to health and well-being of older people, respectively.

Secondly, we chose to focus on the terms, co-creation, co-production, and co-design in our search for relevant literature to capture the latest literature regarding citizen participation. Including previously used concepts such as collaboration, citizen engagement, user involvement, and public participation in our search might have captured other relevant and historical literature on citizen participation among older people.

Thirdly, we searched for two relatively extensive concepts, co-creation and co-production and health and well-being, which made it difficult to identify the most suitable databases for the searches. The concepts of co-creation and co-production are rooted in economy and public administration, whereas the concepts of health and well-being are traditionally rooted in natural and social science. We searched ten scientific databases of social sciences and health in which we expected to find literature that linked the concepts of co-creation or co-production to health and well-being. As we were aware that relevant literature connecting the two concepts could be sparse, we decided to search for unpublished literature as well. However, this search only provided one additional relevant paper. Future searches for peer-reviewed scientific literature connecting the concepts of co-creation and co-production to health and well-being should emphasize choosing only relevant databases depending on the specific focus of the search.

## 5. Conclusions

Consistent with our objective, this review provided an overview of literature describing co-creative and co-productive activities addressing the health and well-being of older people and the factors influencing the co-creation and co-production processes. As expected from initial searches on the subject, the amount of literature linking the concepts of co-creation and co-production to health and well-being was sparse, despite a thorough search of both scientific and non-scientific databases. This finding confirms our assumption that co-creation and co-production of activities in relation to healthy ageing is an emerging field and thus this scoping review can be considered state of the art and the basis for future research.

We identified three categories describing the co-creation and co-production of activities to promote the health and well-being of older people: (1) the co-creation of social and physical activities, (2) co-creation as a method to develop age-friendly environments, and (3) discussions of healthy and active aging. The outcomes of the activities were primarily overall formulations of health and well-being with only a few studies specifying a concrete outcome. The co-creation and co-production processes were facilitated by factors at the organizational level, such as having a realistic plan and structure of the process, early engagement, recognition of participants’ time and resources, and giving participants ownership and control, while the main barriers for co-creation were related to the recruitment of participants, their time, and the lack of structure of the co-creation process.

Future research should focus on the co-creation of specific areas of interest, such as social and physical activity interventions for specific subgroups, to be able to target both the intervention and the co-creation process of the participants and take into consideration the factors influencing a co-creation relationship and ways to measure the co-creation effect on health and well-being.

## Figures and Tables

**Figure 1 ijerph-19-13043-f001:**
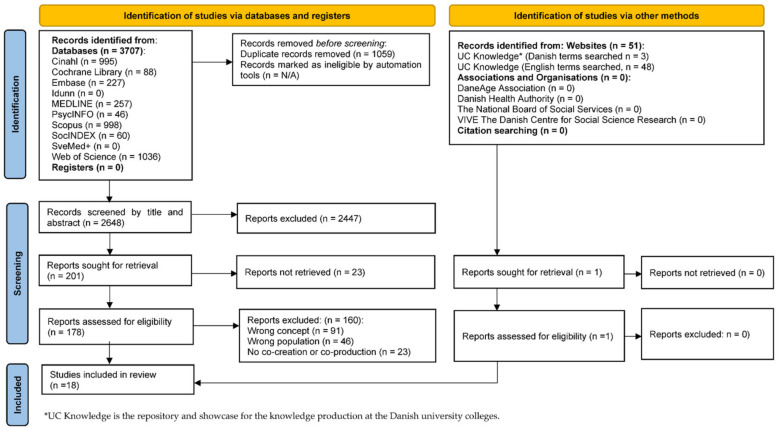
PRISMA flow chart illustrating the study selection process [50].

**Table 1 ijerph-19-13043-t001:** The co-creation and co-production of activities.

Category	*N*	Content
Social and physical activities	7	Leisure activities, music events, and visiting friends [32]; woodwork, furniture restoration, a community garden, school group projects, and cultural activities [34]; designing and weaving a tapestry [36]; rick-shaw bike rides [43]; design projects [48] to community events [49]; and physical activities to reduce sedentary behavior [39].
Development of age-friendly environments	6	Drawing exercises, mapping exercises, model-making, photovoice, and walking interviews were tested and evaluated as ways to design age-friendly homes and neighborhoods [33]; experiential group walks and mapping exercises around the local community to help understand the needs of older people [37] or go along interviews or workshops to identify potentials and barriers for using the local public open space [38,46]; filmmaking to discuss age-friendly cities [47] or co-creation of a framework for a care model to support activities, freedom, and relationships in care homes [35].
Discussions of healthy and active aging	5	Maintenance of health and well-being in old age [40]; development of a typology for being and staying active through old age [41]; identification of older peoples’ needs for aging well [42]; a discussion around an evaluation of a co-creation intervention to reduce sedentary behavior among older people [44]; and discussions of welfare solutions and the importance of collaboration between older people and practice [45].

**Table 2 ijerph-19-13043-t002:** Influential factors for co-creation and co-production.

**Facilitating Factors**		
**Category**	** *N* **	**Content**
General factors	5	Having a realistic scope of the project or having an agreed aim [40,44], interactive tasks and group discussion during the co-creation process to improve communication between stakeholders or emphasized timely communication [44,49], and recognizing participants’ time, competencies, and resources [33,45].
Factors related to participants	7	Early engagement of participants [42,47], recruitment of different stakeholders to ensure representation and legitimacy [40], pairing participants who wanted to engage in co-production and share their experiences [49], engagement of participants in various ways and letting participants manage both the level and the duration of participation [33], letting participants collect and present data to have a sense of control and empowerment [44], and skilled, knowledgeable, and supportive facilitators [33,40].
Tangible factors	3	A familiar, supportive, and accessible location with enough time for the co-creation process to encourage older adults to engage in co-creation [33,40], encourage random seating to facilitate communication and interactions between participants, a flexible structure of workshops in order to let new topics evolve [44], and offering refreshment during the co-creation meetings [33].
**Hindering factors**		
**Category**	** *N* **	**Content**
Recruitment of participants	8	Small number of participants [33,43], selection bias in the recruitment of participants on either side of the co-creation or co-production relationship [38,39,40,42,44,49]; participants were recruited by snowball sampling [38], self-selected [39,42], primarily male [44], or if a higher educational level and status than average population [40,49].
Time and resources	4	Lack of time and resources limited the co-creation events that could take place [33], restricted the access to community members and time available to build trust [37], or hindered replication of the co-creation activity [38]; discussions of how budget constraints could hinder the initiation of a co-creation relationship [43].
Lack of structure of the organization of the work	2	The need for a systematic framework to ensure that co-creation is undertaken in a scientific and reproducible way [44], and an emphasis that informal, minor co-productive activities need more support and coordination [49].

## Data Availability

Not applicable.

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
