# Peer review of "Co-Creation and Co-Production of Health Promoting Activities Addressing Older People—A Scoping Review"

_ijerph, 2022, doi:10.3390/ijerph192013043_

Round 1

Reviewer 1 Report

The article Co-creation and co-production of health-promoting activities addressing older people – a scoping review provides an overview of literature describing co-creative and co-productive activities addressing the health and well-being of older people and the factors influencing the co-creation and co-production processes.

I recommend the manuscript for publication after some major and minor adjustments have been taken into account.

Major revisions:

- Lack of sensitivity analysis of case literature selection process (by the different selecting schemes, or categories related to co-creative and co-productive activities?)

- weak contribution of the research.

Minor comments:

- In the abstract (line 17) the number of studies is 2468, while in line 217 this number is different.

- line 222 – in the statement „Figure I.3.1. Subsection” the „3.1 Subsection” is useless.

- Table 1. I suggest replacing this table with a figure or text. Now, the table is almost empty and the included information is not reader-friendly.  

- lines 152 -153 – the space is useless here.

Reviewer 2 Report

This is a nice paper with added value to the literature. Co-creation and co-production are new buss words, but lack thourough research and strategies, taylor made to specific target groups like older persons.

The paper is well structured, clear written and informative. I only have two issues:

- In your Introduction your rightly and clearly make a difference between co-creation and co-production, it feels like a missed chanche that you combined the two during the paper. Could you please add some more discussion about this distinction (expanding the section in lines 157-171 and 198-203). And perhaps craft recommendations for future work in practice and/or research. For instance, to my experience, co-creation is ex-ante and on group level of end-users, where co-production is with a single end-user during her/his particular service delivery process (you shortly refer to that in lines 76-79 in the Introduction. Have you found evidence for other distinguishing mechanisms, strategies, factors, etc.? Perhaps even reconsider to keep the distinction right from the start and include some results about it (although scarce).

- If I understand the flow chart in figure 1 correctly, the parallel process of identification via other methods resulted in one additional paper. But this 19th paper seems to be kept out of the final 18 studies. Because: 178 reports assessed for eligibility, minus 160 reports excluded, results already in 18 papers in the lower part of the main identification process on the left of figure 1, and the arrow from the right hand side of the figure is emptym. Where is number 19? Please check/confirm. 

Round 2

Reviewer 1 Report

Dear authors, thank you for revising the manuscript and for your responses. I am satisfied with the corrections made to the paper and would like to recommend the study for publication.